# Recent Advances in Microfluidic Devices for Contamination Detection and Quality Inspection of Milk

**DOI:** 10.3390/mi12050558

**Published:** 2021-05-14

**Authors:** Hwee-Yeong Ng, Wen-Chin Lee, Chia-Te Kung, Lung-Chih Li, Chien-Te Lee, Lung-Ming Fu

**Affiliations:** 1Division of Nephrology, Kaohsiung Chang Gung Memorial Hospital, College of Medicine, Chang Gung University, Kaohsiung 833, Taiwan; stan@adm.cgmh.org.tw (H.-Y.N.); leewc@adm.cgmh.org.tw (W.-C.L.); longee01@gmail.com (L.-C.L.); ctlee33@cgmh.org.tw (C.-T.L.); 2Department of Emergency Medicine, Kaohsiung Chang Gung Memorial Hospital, College of Medicine, Chang Gung University, Kaohsiung 833, Taiwan; kungchiate@gmail.com; 3Department of Engineering Science, National Cheng Kung University, Tainan 701, Taiwan

**Keywords:** microfluidic, lab-on-a-chip, lab-on-paper, milk, contamination

## Abstract

Milk is a necessity for human life. However, it is susceptible to contamination and adulteration. Microfluidic analysis devices have attracted significant attention for the high-throughput quality inspection and contaminant analysis of milk samples in recent years. This review describes the major proposals presented in the literature for the pretreatment, contaminant detection, and quality inspection of milk samples using microfluidic lab-on-a-chip and lab-on-paper platforms in the past five years. The review focuses on the sample separation, sample extraction, and sample preconcentration/amplification steps of the pretreatment process and the determination of aflatoxins, antibiotics, drugs, melamine, and foodborne pathogens in the detection process. Recent proposals for the general quality inspection of milk samples, including the viscosity and presence of adulteration, are also discussed. The review concludes with a brief perspective on the challenges facing the future development of microfluidic devices for the analysis of milk samples in the coming years.

## 1. Introduction

Milk is one of the main sources of nutrients necessary for human growth and maintenance of health. As such, it is the most important dietary product for all age groups. However, milk is also a major transmission source for contaminants, such as pathogens, antibiotics, heavy metals, pesticides, and drugs, which can cause various adverse physiological changes and have a serious impact on human health, particularly in infants and young children [1]. Consequently, effective methods for detecting the presence of contaminants, or adulterants, are essential. The quality and safety of milk are generally tested at the downstream processing plant using high-precision instruments and systems. However, while these systems are highly reliable, they tend to be expensive and bulky, such as automatic milk composition analyzer, automatic real-time microbial counting analyzer, etc. Moreover, they require the involvement of professional technical staff. Consequently, the development of microfluidic analysis devices capable of performing the testing process on-site at the point of production has attracted significant interest in recent years.

Microfluidic technology has advanced rapidly over the past decade and has resulted in the development of many lab-on-a-chip [2,3,4,5,6,7,8] and lab-on-paper [9,10,11,12,13,14,15,16] platforms for total analysis systems in fields such as medicine and healthcare, environmental monitoring, cell biology, food safety inspection, and so on. Both platforms offer the advantages of cost-effectiveness, a simple operation, high reliability, high sensitivity, low sample/reagent consumption, and a near-instant on-site detection capability. However, in terms of the transport theory and driving mechanisms used by the two systems, they are very different. In particular, the substrates in lab-on-a-chip devices are generally fabricated of glass, PMMA (poly(methyl methacrylate)), and PDMS (polydimethylsiloxane), and are patterned using photolithography, etching, or laser ablation techniques. Moreover, the working fluid is transported by externally produced pressure or electrical forces [17,18,19,20]. By contrast, in lab-on-paper devices, the substrates are fabricated of paper, and hydrophilic microchannels are defined by printing, wax dipping, cutting, photolithography, or spraying techniques. Furthermore, the sample and buffer solutions flow naturally within the microchannels under the effects of capillary action alone, without the need for any external driving source [21,22,23,24].

The most important purpose of developing microfluidic lab-on-a-chip or lab-on-paper devices is to provide end users with rapid, accurate, easy-to-operate, and cost-effective on-site analysis tools. However, the selection and integration of appropriate detection systems must also be carefully considered. In practice, the choice of detection system depends largely on the limit of detection (LOD) requirements of the target application. For example, the detection of heavy metal residues in milk requires high-sensitivity detection with a LOD on the order of ppb. Such applications generally require the use of high-resolution detection systems such as fluorescence, electrochemistry (EC), electrophoresis (EP), surface-enhanced Raman spectroscopy (SERS), and so on [25,26,27]. However, many alternative quantitative detection systems have also been proposed in recent years, including colorimetry, spectrometry, chemiluminescence (CL), photoelectrochemistry (PEC), and electrochemiluminescence (ECL) [25,28,29]. In general, all of these methods provide a reliable, simple, miniaturized, easily integrated, and cost-effective solution for analyte detection and determination in many different application areas.

This review describes developments in the microfluidic platform technology field for the analysis of milk samples in the past five years. The review commences by describing the methods proposed for sample pretreatment in microfluidic devices, including the separation, sample extraction, and amplification steps. An in-depth review of the latest advances in the use of microfluidic platforms for the detection of aflatoxins, antibiotics, drugs, melamine, and foodborne pathogens in milk samples is then presented. Recent proposals for milk quality inspection and adulterant detection are then briefly described. The review concludes with a brief perspective on the challenges facing the microfluidic platform development field in the dairy business over the coming years, together with the associated opportunities for future research.

## 2. Sample Pretreatment of Microfluidic Devices

The difficulty of sample pretreatment/preparation in food testing is much higher than that for human biological samples. For example, whole blood analysis and diagnosis tests usually require only centrifugation to separate the blood cells and plasma for detection [30]. By contrast, when detecting acidic preservatives in food samples, for example, it is necessary to homogenize the sample first and then extract the test substance by steam distillation before detection can be performed [31,32]. As a result, sample processing in traditional food detection methods usually involves a high cost, a long process, and the use of multiple equipment. This section therefore reviews the pretreatment methods proposed in the literature for milk sample analysis using compact and low-cost microfluidic devices. Broadly speaking, pretreatment methods on microfluidic platforms fall into three main categories, namely separation, extraction, and concentration/amplification.

### 2.1. Sample Separation Microfluidic Devices

In food testing, the sample usually has a complex composition consisting of fats, proteins, and many other components. As a result, it is generally necessary to separate the sample prior to the detection process in order to avoid possible interference with the detection of the target analyte. Singh et al. [33] presented a hand-held microfluidic device incorporating a gold-coated glass slide for the separation of microbes from model solutions and complex matrices. The feasibility of the proposed device was demonstrated by capturing the microbial cells in a yoghurt sample. The results showed that the device captured up to 99.3% of the cells within 60 min. Many other researchers have presented microfluidic platforms for the separation of milk samples in recent years, including antibiotics and toxins [34], quinolone drugs [35], and foodborne bacteria [36]. For instance, Jung et al. [36] developed a magnetophoresis-based microfluidic device for the rapid separation and concentration of foodborne bacteria consisting of a polyethylene tube wrapped around a magnet (see Figure 1a). It was shown that an efficient sample separation performance could be achieved even under high flow rates simply by increasing the length of the tube. For example, when applied to the analysis of milk and homogenized cabbage samples, the device showed a separation efficiency of more than 92% at a flow rate of 40 mL/h.

### 2.2. Sample Extraction Microfluidic Devices

In developing microfluidic devices for the analysis of food samples, one of the key requirements is that of developing miniaturized extraction techniques compatible with lab-on-a-chip and lab-on-paper platforms. The literature contains many proposals for the extraction of various substances in milk samples, including antibiotics [37] and foodborne pathogens [38,39,40,41]. For example, Park et al. [39] presented an integrated rotary microfluidic system for performing the extraction and amplification of foodborne pathogens. As shown in Figure 1b, the system was fabricated using PC (polycarbonate) and PMMA substrates and was designed to perform three functions, namely solid-phase DNA extraction, loop-mediated isothermal amplification (LAMP), and lateral flow strip paper detection. In the proposed device, the extraction step was performed using a solid phase DNA extraction method based on the use of silica bead as a solid matrix. The experimental results showed that the proposed platform was capable of detecting *Salmonella (S.) Typhimurium* and *Vibrio (V.) parahaemolyticus* bacteria in contaminated milk samples with a LOD of 50 CFU (colony-forming unit) within 80 min.

### 2.3. Sample Amplification Microfluidic Devices

In food analysis samples, the target analyte is typically present in only very small quantities. Consequently, some form of sample amplification system must be employed in order to improve the LOD performance. Several microfluidic platforms with integrated amplification systems have been proposed for the detection and analysis of milk based on polymerase chain reaction (PCR) [42,43,44], LAMP [45,46], hybridization chain reaction (HCR) [47], and catalyzed hairpin assembly (CHA) [48]. For example, Li et al. [47] presented an HCR amplification microfluidic aptasensor for the detection of *E. coli* O157:H7 in milk samples. As shown in Figure 1c, an HCR initiator and aptamer were deposited on the substrate, and the stronger binding ability of the pathogen with the aptamer was then exploited to dissociate the aptamer from the initiator and trigger a series of hybridization events to achieve sample amplification. It was shown that the aptasensor had good specificity for the detection of *E. coli* O157:H7 and amplified the signal by around 100 times compared to that of the original (non-amplified) sample.

## 3. Microfluidic Platforms for Milk Sample Analysis

Generally speaking, fresh milk consists of 87% water and 13% total solids (including 4% fat, 3.4% protein, 4.8% lactose, and 0.8% minerals). While the exact quantity of the main components may vary depending on the individual animal, lactation stage, breed, age, and health status, the overall composition does not change very much. As described in Section 1, milk provides many of the essential nutrients required to support human development and growth. However, it is prone to contamination and adulteration, and hence strict quality control and inspection measures are required to safeguard human health. This function is traditionally performed using large-scale, professional equipment in a laboratory environment. However, to reduce costs and enable a more immediate detection result, it is desirable to analyze the milk quality at the point of production using simple, low-cost methods. As a result, the development of microfluidic platforms for the detection and analysis of milk samples has attracted significant interest in recent years [49,50,51]. This section of the review describes some of the most significant lab-on-a-chip and lab-on-paper proposals reported in the literature over the past five years for the detection of some of the most common contaminants in milk, namely aflatoxins, antibiotics, drugs, melamine, and foodborne pathogens. The section concludes by describing several other recent microfluidics-based proposals for general milk quality inspection and adulterant detection.

### 3.1. Aflatoxin Analysis

Aflatoxins are toxic carcinogens and mutagens produced by certain fungi or molds. They can pose serious health risks to humans, including developmental delay, liver damage, and liver cancer [1,52]. Consequently, reliable methods for their detection in foods and beverages are urgently required. While there are many different types of aflatoxins naturally occurring in nature, two of the most common in contaminated milk samples include aflatoxin B1 (AFB1; reference values less than 0.5 μg/L) and aflatoxin M1 (AFM1; reference values less than 0.5 μg/L) (European commission regulation (EC) no. 1881, 2006) [1]. The recent literature thus contains many microfluidic lab-on-a-chip and lab-on-paper devices for the detection of AFB1 in milk [53,54,55] and AFM1 [56,57,58,59] in milk samples. For example, Sheini [54] developed a microfluidic lab-on-paper colorimetric aggregation assay sensor array for the detection of five mycotoxins classified into three categories, namely aflatoxins, ochratoxins A (OTA), and zearalenone. As shown in Figure 2a, multiple hydrophilic zones were patterned on the paper substrate using an ink printer and the zones were then coated with gold and silver nanoparticles (AuNPs and AgNPs) and different reducing agents in order to form a sensing array. The color change induced by the presence of mycotoxins was then analyzed in order to determine the corresponding toxin concentration. The experimental results showed that the platform achieved detection limits of 2.7, 7.3, 2.1, 3.3, and 7.0 ng/mL for AFB1, AFG1, AFM1, OTA, and zearalenone, respectively. Furthermore, the detection results obtained for AFM1 in milk samples were shown to be consistent with those obtained using a conventional high-performance liquid chromatography (HPLC) detection method [54]. Soares et al. [55] presented a multiplexed microfluidic lab-on-a-chip fluorescence immunoassay sensor for the rapid detection of AFB1, OTA, and deoxynivalenol (DON) in food samples. The device consisted of a PDMS substrate patterned with a sample injection microchannel (20 μm height) and four microbead-filled microchannels (100 μm height), and an integrated array of thin-film photodiodes. In the detection process, the microbeads served as the binding medium between the antibody and the target analyte and the analyte concentration was determined using a fluorescence signal acquisition method. When applied to the detection of AFB1, OTA, and DON, the corresponding LODs were found to be 1, 3, and 10 ng/mL, respectively.

Aissa et al. [56] developed a microfluidic lab-on-paper aptasensor device for the detection of AFM1 in milk samples based on electrochemical capacitance spectroscopy (ECS). In the proposed device, carbon screen-printed working electrodes were prepared on the substrate surface and the electrodes were modified with silicon nanocomposite to enhance the capacitance signal. Finally, an anti-AFM1 aptamer was attached to the working electrode to provide a sensing function. The device was shown to achieve a linear ECS response over the AFM1 concentration range of 10 to 500 fM and a LOD of 4.53 fM. He et al. [34] presented a microfluidic lab-on-a-chip aptasensor based on magnetic DNA nanostructure probes for kanamycin (KANA), AFM1, and 17β-estradiol (E2) detection in milk samples. The detection process was implemented using Au-Fe3O4 magnetic complex as a probe, and the target captured by the aptamer on the probe was separated magnetically and transported to a microfluidic electrophoresis chip (MEC) for further analysis by a laser-induced fluorescence (LIF) detection system. It was shown that the device achieved LODs of 0.32, 0.95, and 6.8 pg/mL for KANA, AFM1, and E2 respectively, with a response time of less than three minutes. Table 1 briefly reviews several other microfluidic devices proposed in recent studies for the detection of aflatoxin and other contaminants in milk samples.

### 3.2. Melamine Analysis

Melamine is an organic compound with a high nitrogen content and is often added illegally to food in order to increase its apparent protein content. However, melamine is toxic and can cause many serious health issues, including reproduction damage, bladder infection, kidney stones, acute kidney failure, bladder cancer, and infant death [60]. The detection of melamine in milk samples (non-detectable) requires an extremely high resolution, and hence recent microfluidic devices for melamine analysis generally use high-sensitivity detection techniques such as electrochemistry (EC) [61,62], colorimetric AuNP sensing [63,64], surface-enhanced Raman scattering (SERS) [65,66], and immunosensor [67,68]. For example, Li et al. [62] developed a microfluidic EC DNA-based sensor platform for continuous, real-time melamine detection in whole milk samples. The EC electrode of the biosensor was modified with nucleobase thymine, which subsequently reacted with the melamine in the milk sample to form a DNA triplex. The resulting conformational change produced a corresponding shift in the EC current, from which the melamine concentration was then inversely derived. The results obtained using spiked milk samples showed that the DNA-based device achieved a melamine LOD performance of 2.5 ppm. Gao et al. [63] presented a microfluidic colorimetric lab-on-paper sensor for the detection of melamine in milk samples based on Triton X-100-modified AuNPs. As shown in Figure 2b, Triton X-100 was added to citrate-capped AuNPs to produce a dispersible and stable Triton X-100-AuNP complex. In the presence of melamine, a ligand exchange occurred, which resulted in the release of some of the Triton X-100 from the AuNPs and the subsequent formation of AuNP agglomerates. The resulting color change was then used to determine the corresponding melamine concentration. The results showed that the device achieved a LOD of just 5.1 nM.

Krafft et al. [66] developed a microfluidic lab-on-a-chip device combined with a SERS detector for the detection of melamine in whole milk samples. As shown in Figure 2c, the device consisted of a nanoporous membrane designed to both enrich the sample through a selective electrokinetic filtering process and facilitate SERS through the use of a AgNP layer on its under surface. The experimental results showed that the device achieved a detection limit of 1 ppm. However, it was speculated that the detection performance could be further improved by on-target sample drying using an additional gas flow. Chen et al. [69] presented a centrifugal microfluidic lab-on-a-chip immunoassay device for the detection of six contaminants’ residues in milk samples, namely chloramphenicol (CAP), tetracycline (TC), enrofloxacin (ENR), cephalexin (CEX), sulfonamides (SAs), and melamine. In the immunoassay process, the mixture in the sample chamber was centrifuged into the reaction chamber (containing antibody solution) and two affinity and binding operations were then performed to complete the immunoassay reaction. It was shown that the device allowed the determination of CAP, TC, ENR, CEX, SAs, and melamine with LODs of 0.92, 1.01, 1.83, 1.14, 1.96, and 7.80 μg/kg, respectively. Furthermore, the total reaction time was less than 17 min. Table 1 briefly reviews several other microfluidic devices proposed in recent studies for the detection of melamine and other contaminants in milk samples.

### 3.3. Antibiotic and Drug Analysis

In large-scale dairy production, the cows are often injected with drugs and antibiotics in order to increase the milk yield. However, residues from these drugs and antibiotics may have a harmful effect on human health. A similar problem can arise in the use of certain animal feeds and environmental medications [70]. Consequently, the literature contains many microfluidic devices for the detection of antibiotics and drugs, such as ampicillin [71], β-lactamase [72,73], 17β-estradiol [74], bacitracin zinc [75], chloramphenicol [76,77], clarithromycin [78], clenbuterol [79], ciprofloxacin [80], enrofloxacin [81], monensin [82], norfloxacin [83], streptomycin [84], and tylosin and tilmicosin [85].

Zhang et al. [73] developed a microfluidic lab-on-paper lateral flow immunoassay device for the detection of β-lactams in milk samples. As shown in Figure 3a, the immunoassay platform used amorphous carbon nanoparticles (ACNs) to label anti-receptor monoclonal antibodies (mAb) to form a complex with the unlabeled β-lactam receptors. The results obtained for the detection of 22 β-lactams showed that the measurement sensitivity achieved using ACNs was around 2 times better than that using the same immunoreagents but with colloidal gold as indirect labeling receptors. Furthermore, the sensitivity was around 10 times better than that obtained without using any labeling at all. Zhang et al. [76] presented a microfluidic MCE aptasensor for the detection of chloramphenicol and kanamycin in milk and fish samples. As shown in Figure 3b, the device used a stir bar coated with AuNPs to facilitate DNA multi-arm junctions recycling (MAJR) for signal transduction and amplification purposes. In the detection process, the target antibiotics reacted specifically with the aptamer probes on the stir bar to produce different single-stranded DNA primers. The primers then triggered the MAJR process to connect the three-arm and four-arm DNA strands of the different targets. Finally, the multi-arm DNA was separated by MCE for quantitative detection. It was shown that the device achieved LODs of 0.52 and 0.41 pg/mL for chloramphenicol and kanamycin, respectively.

Zeng et al. [78] developed a microfluidic lab-on-paper immunochromatographic assay device for the rapid detection of four macrolides in milk samples, namely clarithromycin, erythromycin, roxithromycin, and azithromycin. In performing the detection process, antigen and goat anti-mouse IgG antibody were first coated on an immune paper strip as the test (T) line and control (C) line, respectively. Monoclonal antibody was then labeled with colloidal gold in the detection solution, and sample solution was added to prompt the antibiotics to react with the monoclonal antibody. Finally, the mixed solution consisting of the target antibodies and the monoclonal antibodies was dropped onto the previously prepared immune paper strip. The antibodies and colloidal gold-labeled monoclonal antibodies then underwent a competitive interaction with the immunochromatographic band to produce a color change from which the antibiotic concentration was determined. The microfluidic device was shown to be capable of detecting clarithromycin, erythromycin, roxithromycin, and azithromycin in milk samples with LODs of 0.095, 0.085, 0.055, and 0.175 ng/mL, respectively. Meng et al. [84] presented a microfluidic EC aptasensor for the detection of streptomycin in milk samples. As shown in Figure 3c, AuNPs were first coated on the working electrode, and monolayer thiolated cDNA/aptamer duplex (dsDNA) and 6-mercapto-1-hexanol (MCH) were then fixed on the AuNPs to modify the working electrode (Au/SPCE). Finally, a metal-organic framework (MOF) bio-bar code and Exo I enzyme were used to assist the target cycle for dual-signal amplification and achieve a high-sensitivity streptomycin detection performance. The results showed that the aptasensor provided a good linear response over the streptomycin concentration range of 0.005~150 ng/mL and achieved a LOD of 2.6 pg/mL. Table 2 briefly reviews several other microfluidic devices proposed in recent studies for the detection of antibiotics and drugs in milk samples.

### 3.4. Foodborne Pathogens Analysis

The contamination of milk by pathogens is one of the main causes of human health problems and gastrointestinal infectious diseases. Pathogenic bacteria grow and reproduce rapidly within milk and metabolize to produce all manner of toxins. Although the pathogens in milk are killed in the downstream treatment process, the toxins remain active and can cause serious health issues if consumed. Among the many pathogenic bacteria, *Staphylococcus (S.) aureus, Streptococcus, Escherichia (E.) coli, Listeria, Salmonella, Vibrio (V.) vulnificus*, and *Bacillus anthracis* are among the most common [86]. Many of these bacteria are pathogens of bovine mastitis, one of the most widespread and economically costly diseases in the dairy industry. Traditional detection methods for foodborne bacteria involve sample culturing using an appropriate medium. However, while these methods have a high sensitivity, a low cost, and good accuracy, they are time-consuming, laborious, and must be performed in a laboratory setting. To address these problems, various alternative techniques for pathogen detection in milk have been proposed in recent years, including PCR, LAMP, enzyme-linked immunosorbent assay (ELISA), and microarray-based systems [87,88,89]. However, these methods still require the use of laboratory equipment and skilled technical personnel. Thus, the problem of developing microfluidic platforms for the on-site detection of common foodborne pathogens in milk samples has attracted great interest in the recent literature. Typically, these platforms have been designed for the detection of *E. coli* [86,90,91,92,93], *S. aureus* [94,95,96], *Listeria* [97,98,99], and *Salmonella* [100,101,102,103,104].

*E. coli* is a Gram-negative bacterium which lives mainly in the human intestine. In mild cases, it can cause symptoms such as diarrhea, gastroenteritis, inflammation, and malnutrition. However, in severe cases, it can cause more serious problems such as sepsis and hemolytic uremic syndrome [105]. Various microfluidic platforms have been developed for the detection of *E. coli* in milk samples, including lab-on-a-chip devices combined with immunoassay [78], aptamer and fluorescence detection [76], rolling circle amplification (RCA), and fluorescence detection [91], and lab-on-paper devices based on RCA and colorimetric detection [105], colorimetric detection [86,106], and immunoassay [90,92,93]. Sun et al. [105] developed a folding microfluidic lab-on-paper RCA device with DNAzyme-containing DNA superstructures for *E. coli* detection in juice and milk samples. As shown in Figure 4a, the device first extracted the protein molecules from the cells in the sample and then used a DNA superstructure to immobilize the RNA and produce a target-induced cleavage of the DNAzymes. Finally, isothermal RCA was performed to replicate the *E. coli* pathogens. The experimental results showed that the proposed device allowed the detection of *E. coli* K12 with a LOD of 103 CFU/mL within 35 min. Zhang et al. [107] presented a microfluidic MCE device based on a specific aptamer binding strategy for detecting the concentration of *E. coli* in drinking water and milk samples. In the proposed device, the target pathogen initially combined with SYBR gold-labeled aptamer to form a complex. The free aptamer was then separated from this complex by capillary electrophoresis and detected using a LIF technique (see Figure 4b). The LOD of the microfluidic device was shown to be 3.7 × 10^2^ CFU/mL.

*S. aureus* is a Gram-positive bacterium associated with various gastrointestinal infections and dermatitis. The literature contains various proposals for determining the *S. aureus* concentration in milk samples based on techniques such as optical cell counting, colorimetric detection, EC biosensing, and immunoassays [94,95,96]. For example, Eissa and Zourob [95] developed a microfluidic dual EC/colorimetric platform for the detection of *S. aureus* in milk and water samples based on a gold electrode sensor coated with a specific peptide sequence coupled with magnetic nanoparticles. In the as-prepared condition, the electrode surface had a black appearance. However, the subsequent reaction between the peptide sequence and the *S. aureus* protease solution prompted the release of some of the magnetic nanoparticles, causing a partial uncovering of the gold electrode surface. The *S. aureus* concentration was then determined either qualitatively based on a simple visual inspection of the electrode color by the naked eye, or more precisely by means of square wave voltammetry using an iron/ferricyanide redox pair as the EC detection signal. The LOD of the latter approach was determined to be 3 CFU/mL with a response time of 1 min.

*Listeria*, also known as *Listeria (L.) monocytogenes*, is a pathogen of listeriosis and is one of the deadliest foodborne pathogens. *Listeria* usually uses food as a vector of infection and can cause many serious diseases, including meningitis, sepsis, and miscarriage [99]. Various microfluidic platforms have been proposed for detecting *Listeria* in milk samples using such techniques as chemiluminescence, colorimetry, immunoassays, EC biosensors, and PCR amplification combined with LIF [97,98,99]. For example, Silva et al. [98] developed a microfluidic EC immunosensor device for the detection of *Listeria* p60 proteins. In the proposed device, a sandwich immune pair consisting of antibodies for the target protein and Listeria spp. p60 proteins was immobilized on the surface of the electrode and blocked with BSA. The reaction process was then facilitated using an additional secondary antibody and alkaline phosphatase (ALP) with 3-indoxyl phosphate/silver ions. Finally, the concentration of p60 proteins was determined via the voltametric stripping of the enzymatically deposited silver. The LOD of the microfluidic device was found to be 1.5 ng/mL under optimal conditions with a response time of less than 3 h.

*Salmonella* is a foodborne pathogen with symptoms of dehydration, fever, diarrhea, and stomach cramps within 6–72 h of consumption [108]. Many microfluidic devices have been proposed for the detection of *Salmonella* in milk samples. Generally speaking, these devices use colorimetric, immunoassay (microchip-based, thread-based, rotary paper-based), EC biosensing, fluorescence, and PCR amplification combined with LIF [100,101,102,103,104]. For example, An et al. [108] presented a microfluidic lab-on-a-chip platform in which a microfluidic droplet device was used to encapsulate single *Salmonella* cells in a growth medium and the encapsulated droplets were then collected. Finally, the Salmonella concentration was determined by counting the positive sample droplets using a fluorescence detection technique (see Figure 4c). The experimental results obtained using spiked milk samples showed that the device achieved a LOD of 50 CFU/mL. Choi et al. [109] developed a microfluidic thread-based immunoassay device for the detection of *Salmonella* enterica serotype Enteritidis in food samples. As shown in Figure 4d, the hydrophilicity of the cotton thread was enhanced by using polysiloxane as a pore modifier and antibody-gold nanoparticles (dAb-AuNPs) and anti-mouse IgG were then coated on thread-based strips as the test zone and control zone, respectively. The interaction between the target antigen and the antigen-dAb-AuNPs produced complexes which subsequently bound to the captured antibodies and generated a detection signal with an intensity proportional to the Salmonella concentration. The results showed that the intensity of the signal obtained using the proposed device was around 10 times higher than that produced by an equivalent device without polysiloxane modification. Moreover, the LOD was of the order of 100 CFU/mL. Table 2 briefly reviews several other microfluidic devices proposed in recent studies for the detection of foodborne pathogens in milk samples.

### 3.5. Other Analysis and Applications

In addition to the analytes described above, microfluidic devices have also been proposed for the testing of various other contaminants or adulterants in milk, including alkaline phosphatase (ALP) [110], preservative H_2_O_2_ [111], and heavy metals [69,112]. For example, Mahato and Chandra [110] developed a microfluidic lab-on-paper immunosensor device for the detection of ALP, in which the immunosensor probe was covalently immobilized on the surface of the functionalized paper device by ALP antibody and the interaction between the probe and the ALP produced a blue-green precipitate immune complex as an analysis signal (see Figure 5a). The ALP concentration was then determined either qualitatively by the naked eye or quantitatively via colorimetric detection. The linear range of the microfluidic device for ALP detection was shown to be 10–1000 U/mL with a LOD of 0.87 U/mL. Khoshbin et al. [112] presented a microfluidic lab-on-paper array aptasensor device for the simultaneous detection of Hg^2+^ and Ag^+^ in human serum, water and milk samples. As shown in Figure 5b, the device was implanted with fluorescent aptamer and ion detection was performed by measuring the fluorescence quenching intensity induced by the Förster Resonance Energy Transfer (FRET) effect in the presence of the target analyte. The results showed that the platform achieved LODs of 1.33 and 1.01 pM for Hg^2+^ and Ag^+^, respectively.

Milk quality inspection generally involves checking the milk fat, milk protein (lactoferrin), lactose, total solids, somatic cell count, viscosity, and specific DNA sequences. Various microfluidic devices have been proposed for evaluating the lactoferrin [113], lactose [114], somatic cell count [115], adulterants [116,117] viscosity [118], and specific DNA sequences [119] of milk samples. For example, Wang et al. [113] developed a lab-on-paper immunoassay device for the detection of bovine lactoferrin in which a microfluidic immunoassay strip was coated with lactoferrin antigen as the test line and anti-mouse IgG as the control line (see Figure 5c). In the detection process, the correlated antigen and colloidal gold-labeled anti-lactoferrin monoclonal antibody served as competitors in the presence of hybridoma cells secreting IgG in an ELISA assay. The linear detection range for lactoferrin in milk samples was found to be 9.76–625 ng/mL with a LOD of 0.01 ng/mL. Fan et al. [116] developed a milk carton with an integrated microfluidic lab-on-paper colorimetric device for rapid milk quality testing. As shown in Figure 5d, the microfluidic device was printed on the outer surface of the milk carton and was coated with a detection reagent in the reaction zone. In the detection process, a drop of milk was dropped onto the sample zone, and the milk flowed to the reaction zone under the effects of capillary action and prompted a colorimetric reaction. It was shown that the microfluidic device enabled the simultaneous detection of urea, protein, and nitrite with concentrations as low as 0.1, 16, and 0.1 mg/mL, respectively. Table 2 briefly reviews several other microfluidic devices proposed in recent studies for the detection of preservatives and other common adulterants in milk samples.

## 4. Conclusions

Compared with traditional laboratory systems, microfluidic platforms increase the processing speed, reduce the cost, and improve the sensitivity, accuracy, and repeatability of detection in real samples. Furthermore, with their small size and straightforward operation, they can be easily deployed on-site without the need for skilled technical operators. As a result, they have significant potential for many sensing and detection applications in the medicine, environmental monitoring, and food processing and safety monitoring fields. This review has described the major advances in microfluidic lab-on-a-chip and lab-on-paper devices for the pretreatment, contaminant analysis, and quality inspection of milk samples in the past five years.

In general, the review has confirmed that microfluidic devices have significant potential for on-site application in the dairy industry. Many significant devices and platforms have been proposed. The review has shown that microfluidic devices integrated with amplification components such as PCR [39], LAMP [45], HCR [47], CHA [48], and RCA [91] provide an effective means of performing the timely detection of foodborne pathogens, such as S. aureus, Streptococcus, E. coli, Listeria, Salmonella, V. vulnificus, and Bacillus anthracis in milk samples. Integrated rotating microfluidic systems with multiple functions such as DNA extraction and LAMP amplification have also been successfully applied for foodborne pathogen detection [39]. Likewise, a stir bar coated with AuNPs and combined with a microchip device has been shown to be effective in restoring DNA multi-arm link recovery for signal transduction and amplification purposes in the rapid detection of antibiotics [64]. In addition, some microfluidic pre-concentration/amplification techniques have been applied to biological samples such as isotachophoresis (ITP) [120,121,122,123,124,125,126], ion concentration polarization (ICP) [127,128,129,130,131,132], electrokinetic stacking (EKS) [133,134,135,136,137], field amplification stacking (FAS) [138,139,140,141], and multiplex electroanalytical techniques [142,143]. Although it has not been applied to milk analysis, it is also an important reference for future research. The literature contains many other microfluidic lab-on-paper devices for the detection of foodborne pathogens based on immuno-magnetic separation [100,144], droplet encapsulation [108,145,146], and so on. Microfluidic platforms have also been proposed for many other common contaminants and adulterants in milk samples. The results suggest that many of these devices have significant promise for further development and commercialization in the dairy industry.

However, while microfluidic devices have advanced rapidly in recent years and have found growing use in a wide range of on-site detection applications nowadays, there are still several major challenges to be overcome. One of the most pressing challenges is that of the pretreatment of food samples. For example, in performing milk sample detection, it is generally necessary to perform pretreatment steps such as extraction, separation, distillation, amplification, or pre-concentration prior to the detection process itself in order to obtain a purified or enhanced-detection target matrix. In most current microfluidic devices, the pretreatment process is performed using some form of external equipment prior to testing. However, this not only increases the cost and complexity of the detection process, but also increases the detection time and raises the need for additional processing equipment. Thus, the potential of microfluidic devices for on-site timely detection is greatly impaired. Therefore, the need for a pretreatment technology that can be directly integrated with a microfluidic platform or the development of a microfluidic-based pretreatment device is still an important issue. Furthermore, many of the microfluidic devices proposed in the recent literature rely on the use of high-sensitivity detection equipment such as mass spectrometers, electrochemical instruments, fluorescence spectrometers, and Raman analyzers. These microfluidic platforms, combined with high-resolution analytical instruments, can reach specifications in terms of quantification, detection limits, and error ranges. However, these high-resolution analytical instruments still have the disadvantages of being expensive, large in size, requiring professional operation, and not easy to detect on-site. In addition, there are still some disadvantages, such as the difficulty of multistep integration of fluid flow manipulation and the easy clogging of microchannels, and these devices have not yet passed routine use tests [147]. Replacing these instruments with low-cost alternatives, preferably directly integrable with the microfluidic platforms themselves, is also a challenging but highly important concern. Finally, while microfluidic devices have been developed for many common contaminants and adulterants in milk samples, many other analytes have yet to be considered, including calcium, ammonia, chloride, e-fructosyl-lisine, free fatty acids, L-lactic acid, milk urea nitrogen, peroxidase, and preservatives.

In conclusion, this article has reviewed the major developments in the microfluidics field for the analysis of milk and dairy products in the past five years. It is hoped that the analysis and perspectives provided in this review may spur the further development of microfluidic devices in the dairy industry and other food safety testing and monitoring fields and contribute to human health and general wellbeing as a result.

## Figures and Tables

**Figure 1 micromachines-12-00558-f001:**
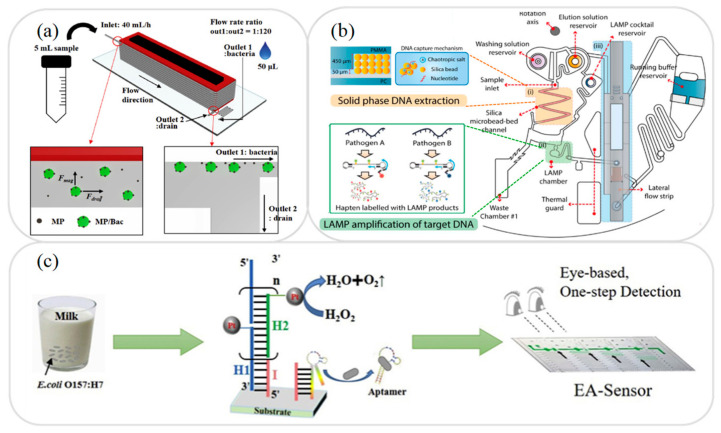
(**a**) Schematic illustration showing working principle of magnetophoresis-based microfluidic device for sample separation and concentration during pretreatment process. Reprinted with permission from ref. [36]. Copyright 2020 Elsevier. (**b**) Schematic illustration showing working principle of extraction and amplification of DNA on microfluidic device. Reprinted with permission from ref. [39]. Copyright 2017 Elsevier. (**c**) Schematic illustration showing HCR amplification of bacteria on microfluidic device. Reprinted with permission from ref. [47]. Copyright 2020 Elsevier.

**Figure 2 micromachines-12-00558-f002:**
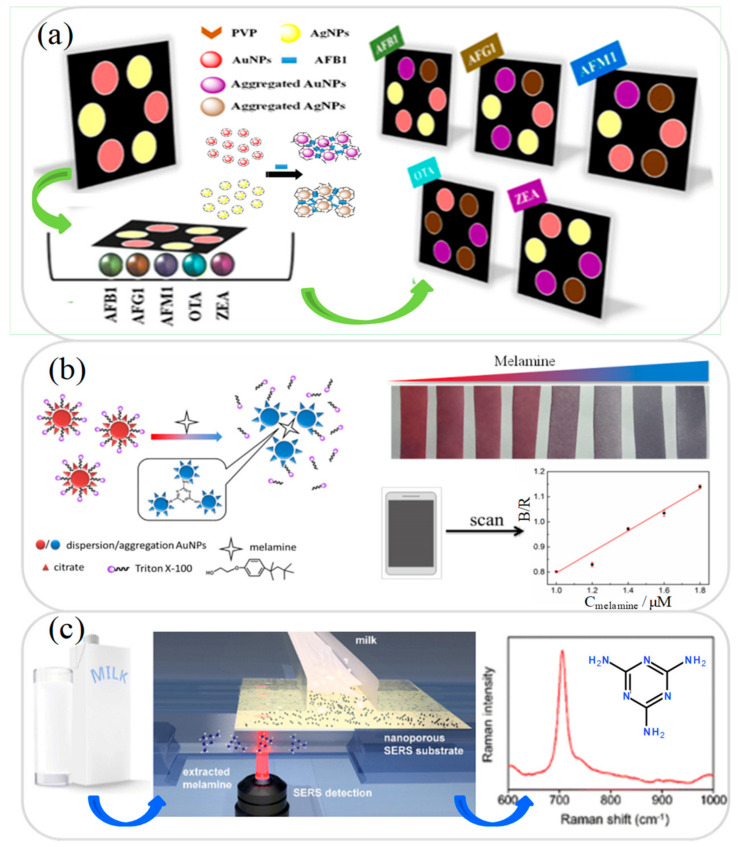
(**a**) Microfluidic colorimetric array device based on AuNPs and AgNPs for detection of aflatoxins, ochratoxins, and zearalenone. Reprinted with permission from ref. [54]. Copyright 2020 Springer. (**b**) Schematic illustration showing working principle of microfluidic colorimetric sensor based on modified AuNPs for detection of melamine. Reprinted with permission from ref. [63]. Copyright 2018 Elsevier. (**c**) Schematic illustration showing microfluidic device combined with SERS detector for determination of melamine. Reprinted with permission from ref. [66]. Copyright 2020 Springer.

**Figure 3 micromachines-12-00558-f003:**
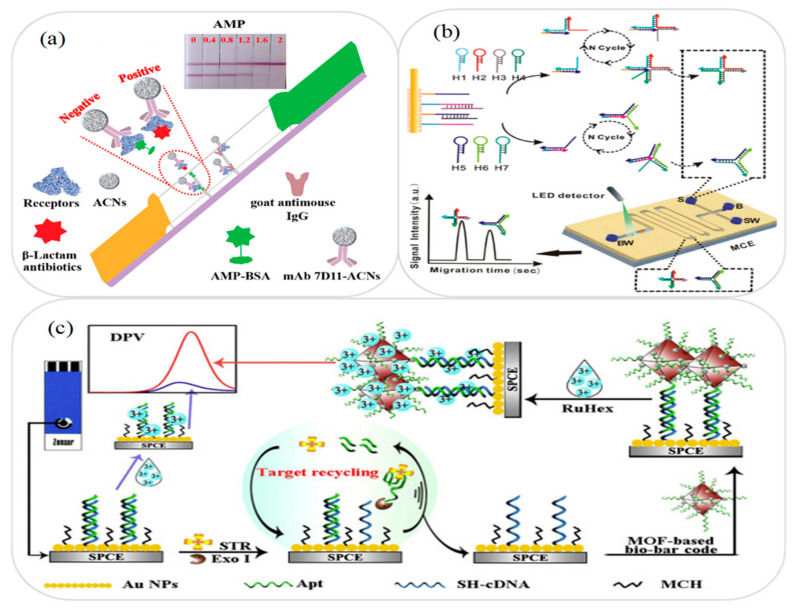
(**a**) Microfluidic lateral flow immunoassay device for detection of 22 β-lactams. Reprinted with permission from ref. [73]. Copyright 2020 Elsevier. (**b**) Schematic illustration showing working principle of MAJR amplification biosensor and microfluidic MCE aptasensor for detection of chloramphenicol and kanamycin. Reprinted with permission from ref. [76]. Copyright 2019 Elsevier. (**c**) Schematic illustration showing working principle of microfluidic EC aptasensor for detection of streptomycin. Reprinted with permission from ref. [84]. Copyright 2020 Elsevier.

**Figure 4 micromachines-12-00558-f004:**
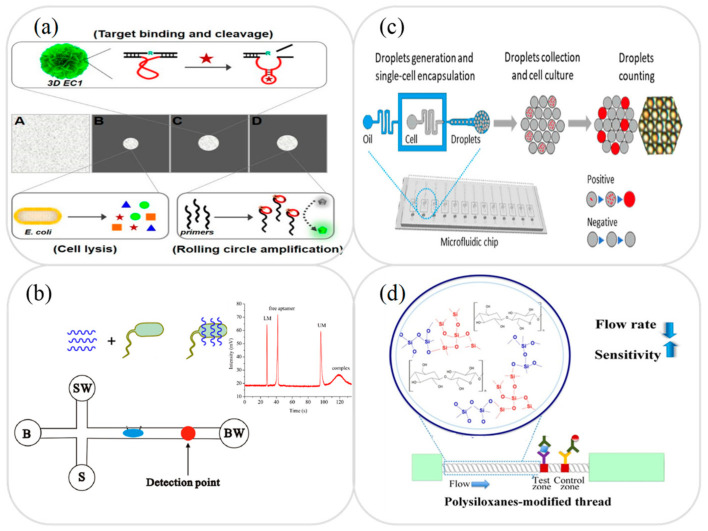
(**a**) Folded microfluidic RCA amplification device for *E. coli* detection. Reprinted with permission from ref. [105]. Copyright 2019 MPDI. (**b**) Schematic illustration showing operating principle of microfluidic MCE device based on specific aptamer binding strategy for *E. coli* detection. Reprinted with permission from ref. [107]. Copyright 2019 Elsevier. (**c**) Microfluidic droplet device for quantitative determination of *Salmonella*. Reprinted with permission from ref. [108]. Copyright 2020 Elsevier. (**d**) Microfluidic thread-based immunoassay device for detection of *Salmonella*. Reprinted with permission from ref. [109]. Copyright 2018 Elsevier.

**Figure 5 micromachines-12-00558-f005:**
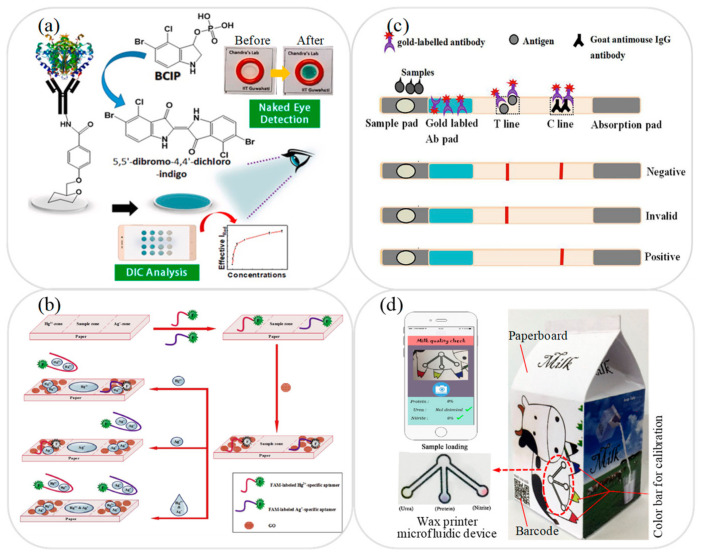
(**a**) Schematic illustration showing working principle of microfluidic immunosensor device for dual naked eye/colorimetric detection of ALP concentration. Reprinted with permission from ref. [110]. Copyright 2019 Elsevier. (**b**) Schematic illustration showing working principle of microfluidic aptasensor device with FRET process for detection of Hg^2+^ and Ag^+^ ions. Reprinted with permission from ref. [112]. Copyright 2020 Elsevier. (**c**) Schematic illustration showing working principle of microfluidic immunoassay device for detection of lactoferrin. Reprinted with permission from ref. [113]. Copyright 2021 Elsevier. (**d**) Schematic illustration and photograph showing milk carton with integrated microfluidic colorimetric device for rapid milk quality testing. Reprinted with permission from ref. [116]. Copyright 2018 Wiley.

**Table 1 micromachines-12-00558-t001:** Summary of the microfluidic devices used for milk contamination analysis.

Device Type, Materials, and Structures	Fabrication Methods	Detection Methods	Target	LOD	Ref.
Lab-on-a-chip, 3-D Plastic	Injection Molding	CM	Aflatoxin M1	0.11 ng/mL	[52]
Lab-on-paper, 2-D Filter paper	Soft Lithography	CM	Aflatoxin B1	10 nM	[53]
Lab-on-paper, 2-D Nitrocellulose MB	Spotting	CM	Aflatoxin M1 Melamine Β-Lactams	0.016 ng/mL 2.5 ng/mL 0.13 ng/mL	[57]
Lab-on-paper, 2-D Filter paper	Soft Lithography	CM	Aflatoxin M1	3 pM @Water 10 nM @milk	[58]
Lab-on-a-chip, 3-D Si/SiO_2_ chip	Deposition	WLRS	Aflatoxin M1	6 pg/mL	[59]
Electrode, 3-D Pt/ZnO/AChE	Coating	EC	Melamine Urea	3 pM, 1 pM	[61]
Lab-on-paper, 3-D Filter paper/PDMS	Cutting and Coating	CM	Melamine	0.1 ppm	[64]
Lab-on-paper, 2-D Filter paper	Dip-coating	SERS	Melamine	1 ppm	[65]
Lab-on-a-Chip, 3-D gold/Quartz	Soft Lithography	Flu	Ag^+^ Hg^2+^	0.038 nM, 0.054 nM	[67]
Lab-on-paper, 2-D Nitrocellulose MB	Spotting	Flu	Aflatoxin M1 Melamine	0.009 ng/mL 0.024 ng/mL	[68]

CM: Colorimetric; EC: Electrochemical; Flu: Fluorescence; MB: Membrane; SERS: Surface-enhanced Raman scattering; WLRS: White light reflectance spectroscopy.

**Table 2 micromachines-12-00558-t002:** Summary of the microfluidic devices used for milk contamination analysis.

Device Type, Materials, and Structures	Fabrication Methods	Detection Methods	Target	LOD	Ref.
Lab-on-paper, 2-D Paper/AgNPs	Printing	EC	Antibiotic	10 μg/mL	[71]
Lab-on-a-chip, 3-D PMMA	Cutting and Adhesive	CM	Β-Lactamase	0.05 mg/mL	[72]
Lab-on-paper, 2-D Millipore MB	Dip-coating	CM	17 Β-Estradiol	0.25 μg/L	[74]
Lab-on-paper, 2-D Nitrocellulose MB	Spotting	CM	Bacitracin Zinc	0.82 ng/mL	[75]
Lab-on-a-Chip, 3-D Gold/Quartz	Soft Lithography and Modify	Flu	CP Kanamycin	0.52 pg/mL 0.41 pg/mL	[76]
Lab-on-paper, 3-D CG paper	Waxing and Coating	CM	Clenbuterol	0.2 ppb	[79]
Electrode, 3-D AuNPs	Coating	EC	Monensin	0.11 ng/mL	[82]
Lab-on-paper, 2-D Filter paper	Waxing and spotting	Flu	Norfloxacin	10 pg/mL	[83]
Lab-on-paper, 2-D Nitrocellulose MB	Spotting	Flu	Tylosin	2 ng/mL	[85]
Lab-on-a-chip, 3-D PDMS	Soft Lithography	Flu	*E. Coli*	80 cells/m	[91]
Lab-on-a-Chip, 3-D Quartz	Soft Lithography	Flu	*S. typhimurium* *P. aeruginosa*	15 CFU/mL 5 CFU/mL	[93]
Lab-on-paper, 3-D chitosan/chondroitin sulfate	Deposited layer-by-layer	EC	*S. aureus*	2.8 CFU /mL	[96]
Lab-on-a-Chip, 3-D cloth-based	Cutting and screen-printing	CL	*L. monocytogenes*	1.1 fM.	[97]
Lab-on-a-Chip, 3-D Quartz	Soft Lithography	Flu	*E. coli* *L. monocytogenes* *S. typhimurium*	2.1 ng/μL 1.8 ng/μL 2.4 ng/μL	[99]
Lab-on-a-Chip, 3-D Ag/AgCl polystyrene	Screen printing and Cutting	EC	*S. typhimurium*	7.7 cells/mL	[101]
Lab-on-a-Chip, 3-D Electrodes	Deposition modified	EC	Salmonella Cells	4 CFU/mL	[103]
Lab-on-paper, 3-D Filter paper/PL	Waxing	CM	*E. coli*	10 CFU/mL	[106]
Lab-on-a-chip, 3-D PLA/PDMS	3D-printing and Soft Lithography	CM	*E. coli* O157:H7	50 CFU/mL	[107]
Lab-on-paper, 2-D Filter paper	Cutting and Dip-coating	CM	H_2_O_2_	1 μM	[111]
Lab-on-a-Chip, 3-D Quartz	Fused	CE analysis	Lactose	2.2 mg/L	[114]

CG: Chromatography; CL: Chemiluminescence; CM: Colorimetric; CP: Chloramphenicol; EC: Electrochemical; Flu: Fluorescence; MB: Membrane; SERS: Surface-enhanced Raman scattering; WLRS: White light reflectance spectroscopy.

## Data Availability

The data presented in this study are available upon request from the corresponding author.

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
