# Peer review of "Recent Advances in Microfluidic Devices for Contamination Detection and Quality Inspection of Milk"

_micromachines, 2021, doi:10.3390/mi12050558_

Round 1
Reviewer 1 Report
The paper presents a good overview of microfluidic methods for determining some contaminants and pathogens in milk. All in all the paper could be useful for those who want to tackle this innovative field and the manuscript may deserve publication, but some changes seem necessary.
Together with the advantages deriving from the use of lab-on-chip and lab-on-paper techniques, the disadvantages should be mentioned since the perfect analytical method does not exist. On lines 513-535 the authors mention something in this sense, but incompletely. For example, what about accurate quantitative analysis and the error range?
Some known disadvantages are already mentioned in literature (DOI: 10.1039/d0fo01278e - DOI: 10.3390/molecules24162869) namely:
- a) fluid flow manipulation for multistep integration, which involves valving and flow velocity control, is a critical parameter and fine control is difficult
- b) The processing required to produce these devices is expensive, complex and requires highly skilled staff
- c) Conventional channel-based microfluidic devices are prone to clogging, and have high labor costs and low productivity, and their “directional pressure” restricts scaling-up capabilities
- d) In general, these are new devices that have not yet been tested for routine use
Line 50: the first time is mentioned an acronym needs to be made explicit. PDMS = Polydimethylsiloxane. The same on line 121 (PC = Polycarbonate, PMMA = poly(methyl methacrylate) and on line 128 (CFU = Colony Forming Unit)
Line 176: It would be useful to specify which legislation is being referred to, since the maximum tolerance limit accepted for AFM1 by the European Union for example is 50 ng/kg in raw milk (Reg EC 1881/2006)
Line 186, "...2.7, 7.3, 2.1, 3.3 and 7.0 ng/mL for AFB1, AFG1, AFM1, OTA and zearalenone, respectively ". Such detection limits seem to be inadequate for control purposes. For example the detection
limit of 2.1 ng/mL for AFM1 means 2100 ng/L = 2.1 µg/L, while the AFM1 reference value for milk is reported to be 0.5 µg/L in milk (line 176)
Line 187-190, "Furthermore, the detection results obtained for AFM1 in milk samples were shown to be consistent with those obtained using a conventional high-performance liquid chromatography (HPLC) detection method". Reference?
Line 221, "...(reference value 0)...". Reference?
Lines 248-250: change the sentence, melamine is not an antibiotic
Line 256: add "and other contaminants" after "melamine"
Lines 263-264: Figure 2c is not present in Ref 66
Line 265: "Summary of" instead of "Summarizes"
Line 287: MAJR means "multi-arm junctions recycling"
Line 367: 3.7×102 CFU/mL. Number 2 should be a superscript character
Line 378: "Qualitatively" instead of "Quantitatively" I think
Line 389: delete CE, it is Electrochemical Immunosensor (perhaps EC?)
Author Response
Reviewer #1:
The paper presents a good overview of microfluidic methods for determining some contaminants and pathogens in milk. All in all the paper could be useful for those who want to tackle this innovative field and the manuscript may deserve publication, but some changes seem necessary.
Together with the advantages deriving from the use of lab-on-chip and lab-on-paper techniques, the disadvantages should be mentioned since the perfect analytical method does not exist. On lines 513-535 the authors mention something in this sense, but incompletely. For example, what about accurate quantitative analysis and the error range?
Some known disadvantages are already mentioned in literature (DOI: 10.1039/d0fo01278e - DOI: 10.3390/molecules24162869) namely:
- a) fluid flow manipulation for multistep integration, which involves valving and flow velocity control, is a critical parameter and fine control is difficult
- b) The processing required to produce these devices is expensive, complex and requires highly skilled staff
- c) Conventional channel-based microfluidic devices are prone to clogging, and have high labor costs and low productivity, and their “directional pressure” restricts scaling-up capabilities
- d) In general, these are new devices that have not yet been tested for routine use
Reply: Thanks to the reviewer. Authors have added a paragraph and a reference in the "Conclusions" section to illustrate the disadvantages of microfluidic technology. (Page 14)
“These microfluidic platforms, combined with high-resolution analytical instruments, can reach specifications in terms of quantification, detection limits and error ranges. However, these high-resolution analytical instruments still have the disadvantages of being expensive, large in size, requiring professional operation and not easy to detect on site. In addition, there are still some disadvantages, such as the difficulty of multistep integration of fluid flow manipulation, the easy clogging of microchannels, and these devices have not yet passed routine use tests [147].
147 Lim, H.; Jafry, A.T.; Lee, J. Fabrication, flow control, and applications of microfluidic paper-based analytical devices. Molecules 2019, 24, 2869.”
Line 50: the first time is mentioned an acronym needs to be made explicit. PDMS = Polydimethylsiloxane. The same on line 121 (PC = Polycarbonate, PMMA = poly(methyl methacrylate) and on line 128 (CFU = Colony Forming Unit)
Reply: Thanks to the reviewer. Authors have annotated these abbreviations. (Pages 2 and 3)
Line 176: It would be useful to specify which legislation is being referred to, since the maximum tolerance limit accepted for AFM1 by the European Union for example is 50 ng/kg in raw milk (Reg EC 1881/2006)
Reply: Thanks to the reviewer. Authors have added “(European commission regulation (EC) no. 1881, 2006)” to the sentence.
Line 186, "...2.7, 7.3, 2.1, 3.3 and 7.0ng/mL for AFB1, AFG1, AFM1, OTA and zearalenone, respectively ". Such detection limits seem to be inadequate for control purposes. For example the detection limit of 2.1 ng/mL for AFM1 means 2100 ng/L = 2.1 µg/L, while the AFM1 reference value for milk is reported to be 0.5 µg/L in milk (line 176)
Reply: Thanks to the reviewer. This article focuses on the simultaneous detection of five contaminations including AFB1, AFG1, AFM1, OTA and zearalenone. The microfluidic lab-on-papedr assay sensor array can be applied to detection of different samples such as pistachio, wheat, coffee and milk. Although the detection limits of some test items cannot meet the specific samples, this study has various items and is worthy of reference.
Line 187-190, "Furthermore, the detection results obtained for AFM1 in milk samples were shown to be consistent with those obtained using a conventional high-performance liquid chromatography (HPLC) detection method". Reference?
Reply: Thanks to the reviewer. Authors have added a reference [54] to the sentence. (Page 5)
Line 221, "...(reference value 0)...". Reference?
Reply: Thanks to the reviewer. Authors have revised the sentence to “(none detectable)”. (Page 5)
Lines 248-250: change the sentence, melamine is not an antibiotic
Reply: Thanks to the reviewer. The sentence has been revised. (Page 6)
Line 256: add "and other contaminants" after "melamine"
Reply: Thanks to the reviewer. The sentence has been revised. (Page 6)
Lines 263-264: Figure 2c is not present in Ref 66
Reply: Thanks to the reviewer. Figure 2c is downloaded from the graphic abstract in the website of Ref 66.
Line 265: "Summary of" instead of "Summarizes"
Reply: Thanks to the reviewer. The typos have been revised. (Pages 7 and 11)
Line 287: MAJR means "multi-arm junctions recycling"
Reply: Thanks to the reviewer. The typo has been revised. (Page 7)
Line 367: 3.7×102 CFU/mL. Number 2 should be a superscript character
Reply: Thanks to the reviewer. The typo has been revised. (Page 9)
Line 378: "Qualitatively" instead of "Quantitatively" I think
Reply: Thanks to the reviewer. The typo has been revised. (Page 9)
Line 389: delete CE, it is Electrochemical Immunosensor (perhaps EC?)
Reply: Thanks to the reviewer. The typo has been revised. (Page 10)
Reviewer 2 Report
This is a well-structured, comprehensive review paper on microfluidic milk analysis devices and techniques. In order for this paper to better guide the future work of this field, the reviewer suggests that the authors discuss relevant, potentially useful (though not already applied to milk analysis) techniques that may solve existing challenges. For example, there are a good number of novel microfluidic platforms for pathogen analysis. The groups of Juan Santiago and Jonathan Posner reported the use of isotachophoresis for integrated DNA extraction and amplification. Jongyoon Han's group reported electrokinetic concentration for integrated DNA extraction and amplification (Angew. Chem. Int. Edit. 2020, 59, 10981–10988) as well as endotoxin detection (Analytical chemistry, 91(3), 2360-2367) that commonly exists in food contamination.
Author Response
Reviewer #2:
This is a well-structured, comprehensive review paper on microfluidic milk analysis devices and techniques. In order for this paper to better guide the future work of this field, the reviewer suggests that the authors discuss relevant, potentially useful (though not already applied to milk analysis) techniques that may solve existing challenges. For example, there are a good number of novel microfluidic platforms for pathogen analysis. The groups of Juan Santiago and Jonathan Posner reported the use of isotachophoresis for integrated DNA extraction and amplification. Jongyoon Han's group reported electrokinetic concentration for integrated DNA extraction and amplification (Angew. Chem. Int. Edit. 2020, 59, 10981–10988) as well as endotoxin detection (Analytical chemistry, 91(3), 2360-2367) that commonly exists in food contamination.
Reply: Thanks to the reviewer. Authors have added a paragraph and some references in the "Conclusions" section to illustrate the application of microfluidic pre-concentration/amplification techniques for biological samples. (Page 14)
“In addition, some microfluidic pre-concentration/amplification techniques have been applied to biological samples such as isotachophoresis (ITP) [120-126], ion concentration polarization (ICP) [127-132], electrokinetic stacking (EKS) [133-137], field amplification stacking (FAS) [138-141] and multiplex electroanalytical techniques [142,143]. Although it has not been applied to milk analysis, it is also an important reference for future research.”
120 Han, C.M.; Catoe, D.; Munro, S.A.; Khnouf, R.; Snyder, M.P.; Santiago, J.G.; Salit, M.L.; Cenik, C. Simultaneous RNA purification and size selection using on-chip isotachophoresis with an ionic spacer. Lab on a Chip 2019, 19, 2741-2749.
126 Sullivan, B.P.; Bender, A.T.; Ngyuen, D.N.; Zhang, J.Y.; Posner, J.D. Nucleic acid sample preparation from whole blood in a paper microfluidic device using isotachophoresis. Journal of Chromatography B 2021, 1163, 122494.
133 Niu, J.; Hu, X.; Ouyang, W.; Chen, Y.; Liu, S.; Han, J.; Liu, L. Femtomolar detection of lipopolysaccharide in injectables and serum samples using aptamer-coupled reduced graphene oxide in a continuous injection-electrostacking biochip. Analytical Chemistry 2019, 91, 2360-2367.
135 Ouyang, W.; Han, J. One-step nucleic acid purification and noise-resistant polymerase chain reaction by electrokinetic concentration for ultralow abundance nucleic acid detection. Angewandte Chemie International Edition 2020, 59, 10981-10988.
